# Modulation of Broiler Intestinal Changes Induced by *Clostridium perfringens* and Deoxynivalenol through Probiotic, Paraprobiotic, and Postbiotic Supplementation

**DOI:** 10.3390/toxins16010046

**Published:** 2024-01-14

**Authors:** Marielen de Souza, Ana Angelita Sampaio Baptista, Maísa Fabiana Menck-Costa, Larissa Justino, Eduardo Micotti da Glória, Gabriel Danilo Shimizu, Camila Rodrigues Ferraz, Waldiceu A. Verri, Filip Van Immerseel, Ana Paula Frederico Rodrigues Loureiro Bracarense

**Affiliations:** 1Laboratory of Animal Pathology (LAP), Department of Preventive Veterinary Medicine, Universidade Estadual de Londrina, Londrina 86057-970, Brazil; marielen.souza@uel.br; 2Laboratory of Avian Medicine (LAM), Department of Preventive Veterinary Medicine, Universidade Estadual de Londrina, Londrina 86057-970, Brazil; anaangelita@uel.br (A.A.S.B.); maisa.menckcosta@uel.br (M.F.M.-C.); larissa.justino@uel.br (L.J.); 3Livestock Gut Health Team (LiGHT), Department of Pathobiology, Pharmacology and Zoological Medicine, Faculty of Veterinary Medicine, Ghent University, 9820 Merelbeke, Belgium; filip.vanimmerseel@ugent.be; 4Biological Science Department, Luiz de Queiroz College of Agriculture, University of São Paulo, Piracicaba 13418-900, Brazil; emgloria@usp.br; 5Department of Statistics, Universidade Estadual de Londrina, Londrina 86057-970, Brazil; shimizu@uel.br; 6Laboratory of Pain, Inflammation, Neuropathy and Cancer, Department of General Pathology, Universidade Estadual de Londrina, Londrina 86057-970, Brazil; camila_ferraz96@hotmail.com (C.R.F.); waverri@uel.br (W.A.V.)

**Keywords:** intestinal health, mycotoxins, detoxification, immunity, necrotic enteritis

## Abstract

Deoxynivalenol (DON) is a predisposing factor for necrotic enteritis. This study aimed to investigate the effects of a DON and *Clostridium perfringens* (CP) challenge on the intestinal morphology, morphometry, oxidative stress, and immune response of broilers. Additionally, we evaluated the potential of a *Lactobacillus* spp. mixture as an approach to mitigate the damage induced by the challenge. One-day-old broiler chickens (*n* = 252) were divided into seven treatment groups: Control, DON, CP, CP + DON, VL (DON + CP + viable *Lactobacillus* spp. mixture), HIL (DON + CP + heat-inactivated *Lactobacillus* spp. mixture), and LCS (DON + CP + *Lactobacillus* spp. mixture culture supernatant). Macroscopic evaluation of the intestines revealed that the CP + DON group exhibited the highest lesion score, while the VL and HIL groups showed the lowest scores. Microscopically, all *Lactobacillus* spp. treatments mitigated the morphological changes induced by the challenge. DON increased levels of reactive oxygen species (ROS) in the jejunum, and CP increased ROS levels in the jejunum and ileum. Notably, the *Lactobacillus* spp. treatments did not improve the antioxidant defense against CP-induced oxidative stress. In summary, a *Lactobacillus* spp. mixture, whether used as a probiotic, paraprobiotic, or postbiotic, exerted a partially protective effect in mitigating most of the intestinal damage induced by DON and CP challenges.

## 1. Introduction

Intestinal health plays a key role in broiler performance and productivity [1]. The growing concern regarding the emergence of multi-resistant bacteria from poultry has led many markets to prohibit or restrict the use of antimicrobials as growth promoters [2,3]. Consequently, maintaining intestinal health in poultry flocks has become increasingly challenging, resulting in the reemergence of intestinal diseases such as necrotic enteritis (NE) [4]. 

Necrotic enteritis (NE) is a bacterial disease caused by strains of *Clostridium perfringens* (CP) [5]. CP is a Gram-positive, anaerobic, spore-forming bacterium that is a natural component of the poultry gut microbiota [6]. This disease occurs when there is an abnormal increase in the population of *C. perfringens* in the gastrointestinal tract (GIT), combined with predisposing factors such as coccidia infection, diets rich in non-starch polysaccharide grains, and exposure to mycotoxins, among others. Virulent strains produce the plasmid-encoded NetB toxin [6,7,8,9]. NE can be manifested as either clinical (resulting in high mortality rates) or subclinical (leading to growth performance failures), with an estimated annual cost of approximately USD 6 billion, equivalent to USD 0.05 per chick [10].

Contamination with mycotoxins is a growing concern in feedstock due to climate change [11]. Mycotoxin exposure significantly contributes to the occurrence of NE, and poultry are frequently exposed to deoxynivalenol (DON) [9,12,13]. DON is one of the most prevalent mycotoxins contaminating finished feed and raw commodities worldwide [14]. In poultry, DON exposure has been linked to villus atrophy, failure in intestinal barrier function, increased intraepithelial lymphocyte infiltration, heightened goblet cell abundance, intestinal oxidative stress, and disruption of gut microbiome diversity [15,16,17,18,19]. Consumption of DON has additionally been linked to increased susceptibility to coccidiosis and necrotic enteritis, particularly in more severe cases [9,12,20].

Probiotics consist of beneficial microorganisms that can enhance host health through various mechanisms [21]. Paraprobiotics, on the other hand, refers to dead probiotic microbial cells and their constituents, making them a preferred choice for immunosuppressed hosts due to the absence of the risk of bacterial translocation [22]. Postbiotics encompass bacterial metabolites, cell-free supernatant (CFS), and soluble factors (products or metabolic byproducts) produced by live bacteria or released following bacterial lysis [23]. *Lactobacillus* spp. strains, in different forms—be it live cells, heat-inactivated cells, or culture supernatants—have demonstrated protective effects in chickens individually challenged with either DON or CP [5,24,25,26,27]. 

This study aimed to investigate the impact of a dual challenge involving DON and CP on intestinal morphology, morphometry, immune response, and oxidative stress in broilers. Additionally, we have evaluated the efficacy of a *Lactobacillus* spp. mixture, administered as a probiotic (live cells), paraprobiotic (heat-inactivated cells), and postbiotic (culture supernatant), as a potential alternative to mitigate the damage induced by both of these factors.

## 2. Results

### 2.1. Effects of DON, C. perfringens, and Lactobacillus spp. Mixture on the Intestines

On the 23rd day, the intestines were macroscopically evaluated (Figure 1). The CP + DON group exhibited a worse intestinal gross appearance than the CP group. Among the *Lactobacillus* spp.-supplemented groups, VL and HIL induced the most effective protective effects. The observed changes included loss of intestinal tonus, hyperemia (Figure 1C), excessive mucus, the presence of yellow peeling content (Figure 1D), and, less frequently, a thick fibrinous mucus layer. 

Intestinal morphometry is directly related to zootechnical performance [28]. In the jejunum, DON, CP, and CP + DON treatments reduced villus height and the villus: crypt ratio compared to the control and *Lactobacillus* spp.-supplemented groups. In the ileum, DON and CP treatments led to a reduction in villus height compared to the other groups, while no significant difference was observed in duodenal morphometry among the experimental groups (Table 1).

As sentinels of the intestinal barrier and local immune response, intraepithelial lymphocytes (IELs) were evaluated [29]. In the jejunum, only DON ingestion did not induce an increase in IELs; however, broilers receiving CP or CP + DON showed increased IEL infiltrate (≈1.5-fold on average for both groups) compared to the control group, while the VL and HIL groups were similar to the control. In the ileum, all treatments induced a higher number of IELs compared to the control (Table 1). The number of goblet cells was evaluated in the ileum, and the *Lactobacillus* spp.-supplemented groups showed a higher abundance compared to the control, DON, CP, and CP + DON treatments.

Morphological alterations induced by the different treatments were evaluated using a lesion score. In the duodenum, CP and CP + DON increased the lesion score by approximately 2.6-fold compared to the control, while the remaining treatments resulted in lesion scores similar to those of the control and DON groups. In the jejunum, all treatment groups showed higher scores than the control group, except for the HIL group. In the ileum, increased lesion scores were observed in all treatment groups except the VL group when compared to the control (Table 1). The most frequent changes observed in the histological evaluation included edema of the lamina propria, lymphocytic infiltrate, cell debris, apical necrosis, adhesion of bacteria on the villi surface, and cytoplasmic vacuolation of enterocytes (Figure 2).

Scanning electron microscopy (Figure 3) was performed to illustrate the effects of *Lactobacillus* spp. mixture, DON, and CP challenge in the jejunum. The control group showed normal villus morphology, while in the DON and CP + DON groups, a thicker mucus layer was observed. The *Lactobacillus* spp. mixture groups showed preserved villi morphology, similar to the control.

### 2.2. Effects of DON, C. perfringens, and Lactobacillus spp. Mixture on Redox Status

Oxidative stress occurs when there is an imbalance between the production of radical species and the antioxidant response in the organism [30]. Our study aimed to investigate the effects of DON and CP on the redox status. To achieve this, samples from the jejunum and ileum were used in lipid peroxidation (TBARS) and superoxide anion production (NBT) assays to assess the oxidative response. Additionally, GSH (reduced glutathione), ABTS (3-ethylbenzothiazoline-6-sulphonic acid), and FRAP (ferric reducing ability) assays were performed to evaluate antioxidant capacity.

In the jejunum, exposure to DON and CP + DON increased TBARS levels compared to the control, CP, and *Lactobacillus* spp.-supplemented groups. An increase in the NBT levels was detected in the CP, CP + DON, and *Lactobacillus* spp.-supplemented groups compared to the control and DON groups (Table 2). DON, CP, and CP + DON challenges reduced GSH levels compared to the control group, with the *Lactobacillus* spp.-supplemented treatments unable to restore GSH levels to those of the control. Regarding FRAP levels, the DON and control groups were similar, while the CP, CP + DON, and *Lactobacillus* spp.-supplemented groups showed higher levels in comparison. There were no significant differences in ABTS levels between the groups.

In the ileum, both CP and CP + DON treatments significantly increased NBT levels by approximately 5.6-fold compared to the control and DON groups. However, the *Lactobacillus* spp.-supplemented treatments were unable to revert superoxide anion production to control group levels. Regarding GSH levels, CP and CP + DON treatments reduced it approximately 2.85-fold compared to the control and DON groups, with the *Lactobacillus* spp.-supplemented treatments showing no significant differences compared to the CP and CP + DON groups. No significant differences were observed among the TBARS, ABTS, and FRAP levels between the groups.

### 2.3. Effects of DON, C. perfringens, and Lactobacillus spp. on Intestinal Secretory IgA Levels

Intestinal secretory IgA levels were assessed at three time points: 7, 14, and 20 days. No significant difference was observed among the treatments (*p* = 0.08). However, with respect to time, a lower IgA concentration was observed at 7 days (1,165,620 ± 636,766 ng/mL), compared to that at 14 (1,488,459 ± 947,397 ng/mL) and 20 (1,610,248 ± 1,050,150 ng/mL) days (*p* = 0.02). No interaction between time and treatment was observed (*p* = 0.20). 

## 3. Discussion

Previous studies have shown that exposure to mycotoxins, such as DON alone or in combination with other mycotoxins like fumonisins, can exacerbate the occurrence of necrotic enteritis (NE) [9,12,13]. In this study, we aimed to investigate the effects of different presentations [probiotic (viable cells), paraprobiotic (heat-inactivated cells), and postbiotic (heat-inactivated cells culture supernatant)] of a *Lactobacillus* spp. mixture on broiler chickens challenged with DON and *Clostridium perfringens* (CP), as previous research has suggested that *Lactobacillus* spp. strains may mitigate the negative effects of single challenge with mycotoxins or CP in the intestine [5,24].

Macroscopic evaluation of the intestines revealed that the CP + DON group exhibited a higher lesion score compared to the group challenged with CP alone, consistent with previous reports indicating that a DON-contaminated diet can increase the severity of NE cases [9,12]. Among the *Lactobacillus* spp. groups, VL and HIL induced the lowest lesion scores. Microscopically, both DON and CP exposure induced intestinal lesions, and the combination of both factors tended to increase the lesion score, although the difference was not significant. The *Lactobacillus* spp. treatments appeared to mitigate the morphological changes induced by both challenges.

Zootechnical performance is closely related to increased intestinal absorptive capacity. Villus height serves as an indicator of the absorption area, and the intestinal crypts are the sites of new enterocyte multiplication [31,32]. The jejunum and ileum were the intestinal segments most affected by both DON and CP challenges, showing a reduction in villus height and the villus: crypt ratio (only in the jejunum) compared to the control and *Lactobacillus* spp.-supplemented groups. Since the small intestine is the primary site of DON absorption, previous studies have reported impaired intestinal morphometry as a consequence of DON ingestion [16,33]. However, in this study, the combination of DON and CP did not worsen the intestinal morphometry compared to single challenge. 

Intraepithelial lymphocytes (IELs) are key components of the intestinal barrier [29]. In this study, exposure to CP increased the number of IELs in both the jejunum and ileum, regardless of DON administration. Similar findings were reported in naturally infected laying hens [34]. However, in the VL and HIL groups, the number of IELs was reduced to control levels in the jejunum but not in the ileum. 

Contrary to previous reports [16,35], DON exposure did not induce an increase in goblet cell density under light microscopy evaluation in this study. However, scanning electron microscopy revealed enhanced mucus presence in the DON-exposed group compared to the control group. *Lactobacillus* spp. supplementation increased the number of goblet cells compared to other groups. Excess mucus can predispose to NE; however, this finding was expected, as the microbiota can influence mucus layer development, and specifically, *Lactobacillus* spp. are known to contribute to strengthening the intestinal mucosal barrier function [36,37,38,39].

Regarding oxidative stress in the jejunum, DON induced lipid peroxidation, and this effect was sustained after the CP challenge. However, the *Lactobacillus* spp. treatments decreased the lipid peroxidation/MDA levels to control levels. Previous research has also reported DON-induced lipid peroxidation [16,40,41], which is associated with mitochondrial damage [42]. The NBT assay quantifies superoxide anions indirectly through their oxidative effects on NBT and is mainly produced by inflammatory cells [43]. CP induced inflammation, as confirmed by higher NBT levels, but the *Lactobacillus* spp. treatments did not reverse this effect. As a consequence of lipid peroxidation and inflammation, lower GSH levels were observed in all DON- and CP-challenged groups compared to control levels. These findings align with other studies that have reported the capacity of DON and CP to reduce intestinal antioxidant defense [16,26,44,45]. The FRAP assay measures tissue ferric reducing ability, and CP-challenged groups showed higher FRAP levels, likely as a response to the inflammatory status and oxidative stress [46].

The ileum is the final segment of the small intestine, and exposure to xenobiotics such as mycotoxins is lower than that in the proximal regions [47]. In this study, the ileum showed no change in the oxidative stress response after DON exposure. It is likely that the levels of mycotoxins were reduced due to the intestinal microbiota’s detoxification activity [48]. Organisms under long-term toxicity might induce adaptations to reduce the damage [49,50,51]. On the other hand, CP induced an inflammatory status resulting in higher levels of NBT and lower levels of GSH, and *Lactobacillus* spp. did not exert a protective effect.

In this study, concomitant exposure to DON and CP did not worsen most of the evaluated parameters compared to single challenge. However, the *Lactobacillus* spp. treatments, especially LV and HIL, were effective in mitigating tissue damage. Viable or heat-inactivated cells from one strain of *L. plantarum* used in this study have a recognized capacity to remove DON [52]. The mechanism of action is still unclear, but based on research with similar microorganisms, it is hypothesized that viable cells can detoxify DON while heat-inactivated cells can bind to the mycotoxin, thereby reducing its toxic effects [53,54,55]. 

The *Lactobacillus* strains used in this experiment underwent in vitro evaluation, revealing their ability to antagonize the growth of *C. perfringens* (inhibition zone on spot on the lawn varying from 14 to 22 mm, data not shown). The protective effects of viable cells against CP-induced damage were likely a result of bacteriocin production targeting *C. perfringens*, whereas heat-inactivated cells might exert a prebiotic effect by modulating the gut microbiome and preventing the proliferation of pathogenic microorganisms [26,56,57,58,59].

## 4. Conclusions

Additional studies are required to clarify the mechanisms of action of the evaluated *Lactobacillus* spp. strains and establish an industrial process for producing and transforming these strains into a commercial product for animal consumption. This is especially crucial as the model employed in this trial (oral gavage) is not applicable in commercial poultry farming. Nonetheless, it can be inferred that these strains exhibited a protective effect, mitigating a significant portion of the intestinal damage induced by DON and *C. perfringens*.

## 5. Materials and Methods

### 5.1. Study Location and Ethical Approval

This study was conducted at the avian medicine experimental facilities at Universidade Estadual de Londrina, Londrina, Paraná, Brazil, and received approval from the institutional ethics committee for the use of animals (*Comitê de Ética no Uso de Animais* CEUA-UEL, protocol number 12433.2018.03, approval date 24 September 2018). 

### 5.2. Animals and Treatments

One-day-old broiler chickens (*n* = 252), Ross 308 lineage, were housed in cages with water, feed, and heating provided, following lineage guidelines [60]. The animals were divided into seven treatment groups (*n* = 36 each), as follows: Control—uncontaminated diet; DON—diet containing DON at 19.3 mg kg^−1^; CP—uncontaminated diet + *Clostridium perfringens* challenge; CP + DON—diet containing DON at 19.3 mg kg^−1^ + *C. perfringens* challenge; VL—diet containing DON at 19.3 mg kg^−1^ + *C. perfringens* challenge + viable *Lactobacillus* spp. mixture; HIL—diet containing DON at 19.3 mg kg^−1^ + *C. perfringens* challenge + heat-inactivated *Lactobacillus* spp. mixture; LCS—diet containing DON at 19.3 mg kg^−1^ + *C. perfringens* challenge + *Lactobacillus* spp. mixture culture supernatant.

### 5.3. Diets 

The experimental diets were formulated to meet the nutritional requirements of the animals (Table 3), consisting of three different diets used during the trial period: Diet 1 (0–6 days)—primarily composed of corn (48.16%) and soybean meal (43.70%), free of DON; Diet 2 (7–14 days)—composed of corn (15%), soybean meal (35.31%), and wheat (40.13%); Diet 3 (15–23 days)—composed of corn (15%), wheat (40.43%), and fish meal (35.31%). Fish meal was added to diet 3 to elevate the crude protein level and create an intestinal environment favorable for the experimental induction of necrotic enteritis [61].

On the 7th day, DON-challenged groups (excluding the control) began receiving a DON-contaminated diet containing 19.3 mg kg^−1^ (Appendix A). The crude DON extract used to contaminate the diets was provided by the Laboratory of Mycology, Luiz de Queiroz College of Agriculture, University of São Paulo. A blend (standard diet + DON) was prepared at the Universidade Estadual de Londrina facilities using a commercial feed mixer. The diets were sent to Lamic laboratory (Santa Maria—RS/Brazil), where the mycotoxins levels were assessed using the HPLC/MS-MS method. Three diet samples were collected throughout the experimental period: first, 0–6 days (diet 1); second, 7–23 uncontaminated diet (diets 2 and 3); and third, 7–23 DON-contaminated diet (diets 2 and 3). The results of the mycotoxin (deoxynivalenol, aflatoxins, fumonisins, and zearalenone) analysis are shown on Table 4.

### 5.4. Necrotic Enteritis Induction

Animals in the CP, CP + DON, VL + DON, HIL + DON, and LCS + DON groups underwent necrotic enteritis induction. They were orally challenged with 4000 oocysts of *Eimeria* spp. from a commercial vaccine (Livaccox^®^, Paulínia, Brazil) and a 10-fold dose of a commercial Gumboro disease vaccine (Bursa F^®^, Campinas, Brazil) [62,63] on the 14th day. The non-challenged groups received 1 mL of sterile PBS to replicate the same stress.

A *C. perfringens* type G, netB positive strain from the Avian Medicine UEL collection was used to challenge the birds. The strain was grown in BHI (Brain Heart Infusion, HiMedia, Sumaré, Brazil) broth at 37 °C for 18 h under anaerobic conditions using a commercial kit (GasPak^®^, Becton Dickinson Osasco, Brazil). From the 16th to the 22nd day, animals received 1 mL of fresh CP culture (approximately 4 × 10^8^ CFU/mL) via oral gavage twice daily (Figure 1). On each challenge day, an inoculum sample was 10-fold diluted and plated on SFP agar^®^ (Becton Dickinson Osasco, Brazil), followed by incubation under anaerobic conditions to determine the CFU count. The non-challenged groups received 1 mL of BHI broth to simulate the same stress.

### 5.5. Lactobacillus spp. Mixture Administration

Animals in the VL, HIL, and LCS groups received 1 mL of a *Lactobacillus* spp. mixture (approximately 2.2 × 10^9^ CFU/mL) via oral gavage every other day throughout the experimental period (Figure 1). Groups not supplemented with the *Lactobacillus* spp. pool received 1 mL of sterile MRS (De Man, Rogosa, and Sharpe medium, HiMedia Sumaré, Brazil) broth. 

The *Lactobacillus* spp. mixture comprised an equal quantity of three strains: two isolated from broiler chickens (*L. reuteri* and *L. plantarum*, not deposited in GenBank) and one from wheat (*L. plantarum*—accession number CP053912) in previous studies [52,64,65]. The strains were grown separately in MRS broth and incubated at 37 °C for 24 h under microaerophilic conditions. Samples were provided in three different forms: (i) a fresh culture of viable *Lactobacillus* spp. mixture; (ii) a heat-inactivated culture of *Lactobacillus* spp. mixture, and (iii) a supernatant culture from a heat-inactivated *Lactobacillus* spp. mixture. The inactivation and mixture preparation followed previously described methods [16,52]. Cell density was assessed daily through 10-fold dilution and plating. 

### 5.6. Sample Collection

Throughout the experimental period, four samplings were conducted. On days 7, 14, and 20, ten animals per treatment group were euthanized, and on day 23, 6 animals per treatment were used for biological sample collection. The intestinal samples underwent macroscopic lesion scoring, histological examination, ELISA (enzyme-linked immunosorbent assay), and oxidative stress response assessments.

Macroscopic intestinal lesion score

On the 23rd day, 6 animals per treatment were euthanized, and an intestinal lesion score was determined following previously described criteria, ranging from 0 to 5 [66].

ELISA

Intestinal fluid was collected from 10 animals per treatment at 7, 14, and 20 days. For this purpose, 2 mL of a wash buffer (PBS pH 7.2, thimerosal 0.01%, 1% BSA, 1 mM phenylmethylsulfonyl fluoride, and 5 mM EDTA) was injected into the proximal duodenum and collected at the distal ileum. The collected samples were then centrifuged at 1200× *g* for 15 min at 4 °C, and the resulting supernatant was collected and stored at −20 °C. The levels of IgA were determined using the chicken IgA ELISA quantitation kit (Bethyl^®^ Laboratories, Montgomery, TX, USA). The assay was performed in triplicate following the manufacturer’s instructions, with the plates read at 450 nm.

Histology and scanning electron microscopy

Morphological and morphometric evaluations were carried out on the intestines (duodenum, jejunum, and ileum) of 6 animals per treatment on the 23rd day. The Swiss roll technique [67] was used to collect and prepare the samples. The tissues were fixed in a 10% buffered formalin solution and subsequently subjected to routine histological processing. Sections of 5 µm thickness were obtained and stained with hematoxylin-eosin (HE) and Alcian Blue (AB). AB staining was utilized to determine goblet cell density.

Morphometric analysis was performed on 30 randomly selected villi and crypts per slide (180 villi and crypts per treatment) using image analysis software (Motic Image Plus 2.0, Motic Instruments, Richmond, BC, Canada). Measurements of villus height, crypt depth, and villus: crypt ratio were conducted in the duodenum, jejunum, and ileum. The morphological evaluation of the intestines followed the scoring system described by Terciolo, et al. [68], with minor modifications (including additional lesions in the score: inflammatory infiltrate, congestion, bacteria adhered to the villi, presence of *Eimeria* spp., and cell debris). The count of intraepithelial lymphocytes (IELs) was performed on 12 randomly chosen villi per slide (72 villi per treatment), considering IELs positioned above the enterocyte nucleus. Goblet cell density was determined exclusively in the ileum by evaluating 15 random villi per slide (90 villi per treatment).

Scanning electron microscopy was conducted specifically in the jejunum. Samples were collected on the 23rd day, fixed in a 2.5% glutaraldehyde buffered solution (sodium cacodylate solution 0.1 M, pH 7.2) for 24 h, and subsequently washed with sodium cacodylate buffer (0.1 M, pH 7.2). Treatment involved exposure to 1% osmium tetroxide in sodium cacodylate buffer (0.1 M, pH 7.2) for 1 h. Subsequent steps included gradual dehydration in different ethanol concentrations (70, 80, 90, 100%) and drying to the critical point using a CPD 030 critical point dryer (Bal-Tec Union Ltd., Vaduz, Liechtenstein). Following this, tissues were coated with gold (Sputter Coater SDC 050, Bal-Tec Union Ltd., Vaduz, Liechtenstein), and the morphology of the intestinal villi was examined using a scanning electron microscope (FEI Quanta 200, Field Electron and Ion Company, Hillsboro, USA).

Oxidative stress response

The oxidative stress response was evaluated in both the jejunum and ileum. On the 20th day, 4 animals per treatment were euthanized, and samples from these tissues were collected in microtubes and preserved at −80 °C until processing. The antioxidant capacity was assessed by quantification of reduced glutathione (GSH) following the method of Sedlak and Lindsay [69], ferric reducing ability (FRAP), and reduction of 2,2′-azino-bis (3-ethylbenzothiazoline-6-sulfonic acid) (ABTS), as described by Katalinic, et al. [70].

The oxidative response was evaluated using the nitroblue tetrazolium assay (NBT) [71] and the quantification of thiobarbituric acid reactive substances (TBARS) [72,73]. Tissue homogenization was carried out in buffer using a Tissue-Tearor (Bjospec, São Paulo, SP, Brazil). For the FRAP, ABTS, NBT, and TBARS protocols, the buffer consisted of KCL (1.15%) and EDTA (0.02 M) for GSH analysis.

### 5.7. Statistical Analysis

The experimental design was entirely randomized, except for the ELISA analysis, which followed a factorial 3 × 7 design (3 time points and 7 treatments). Each animal was considered one experimental unit. Data analysis was carried out using the free software R^®^ version 3.4.4, and ANOVA was performed using the AgroR package with a significance level of 5%. If the means exhibited statistical significance, the data were submitted to a Scott–Knott multiple comparison test at a 5% significance level. Data that did not meet the assumption of normality of errors were subjected to logarithmic transformation before undergoing ANOVA and the Scott–Knott test.

## Figures and Tables

**Figure 1 toxins-16-00046-f001:**
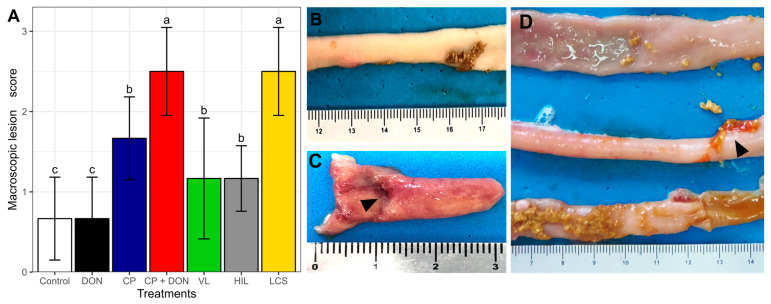
(**A**) Effect of *Lactobacillus* spp. mixture in the macroscopic aspect of the small intestine of broilers challenged with DON and *Clostridium perfringens*. Values are the means ± standard deviation of the mean. ANOVA followed by Scott–Knott multiple comparison test was used to determine statistical differences among groups. ^a,b,c^ Different letters indicate a statistical difference. (**B**) Control—normal gross aspect of intestinal mucosa. (**C**) CP + DON—altered gross aspect of intestinal mucosa, moderate hyperemia, and presence of an ulcer (arrowhead). (**D**) LV—gross aspect of the intestine from the viable *Lactobacillus* spp. mixture group, discrete presence of yellow peeling content (arrowhead), and hyperemia and petechiae are observed. Control—uncontaminated diet. DON (deoxynivalenol)—diet with DON 19.3 mg kg^−1^. CP (*Clostridium perfringens*)—uncontaminated diet + *C. perfringens* challenge. CP + DON—DON 19.3 mg kg^−1^ + *C. perfringens* challenge. LV—DON 19.3 mg kg^−1^ + *C. perfringens* challenge plus viable *Lactobacillus* spp. mixture. HIL—DON 19.3 mg kg^−1^ + *C. perfringens* challenge plus heat-inactivated *Lactobacillus* spp. mixture. LCS—DON 19.3 mg kg^−1^ + *C. perfringens* challenge plus *Lactobacillus* spp. mixture culture supernatant.

**Figure 2 toxins-16-00046-f002:**
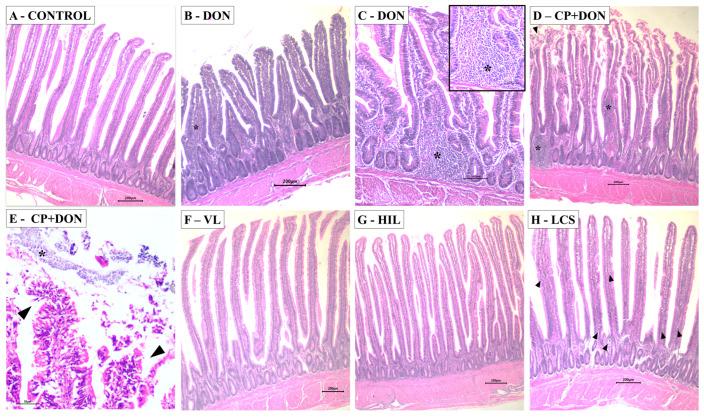
Morphological changes induced by DON, CP, and *Lactobacillus* spp. mixture supplementation on broiler jejunal tissue. (**A**) Control: normal villi morphology. HE, bar 200 µm. (**B**) DON: villi atrophy and enhanced presence of inflammatory infiltrate (*). HE, bar 200 µm. (**C**) DON: enhanced presence of inflammatory infiltrate (*). HE, bar 100 µm. Insert: inflammatory infiltrate predominantly composed by mononuclear cells. HE, bar 50 µm. (**D**) CP + DON: focal area of necrosis (*). HE, bar 200 µm. (**E**) CP + DON: focal area of apical necrosis with myriad of bacterial colonies attached to cell debris (*). HE, bar 50 µm. (**F**) Viable *Lactobacillus* spp. mixture: preserved villi morphology. HE, bar 200 µm. (**G**) Heat-inactivated *Lactobacillus* spp. mixture: preserved villi morphology. HE, bar 200 µm. (**H**) *Lactobacillus* spp. mixture culture supernatant: preserved villi morphology with different development stages of *Eimeria* spp. oocysts (▲). HE, bar 200 µm.

**Figure 3 toxins-16-00046-f003:**
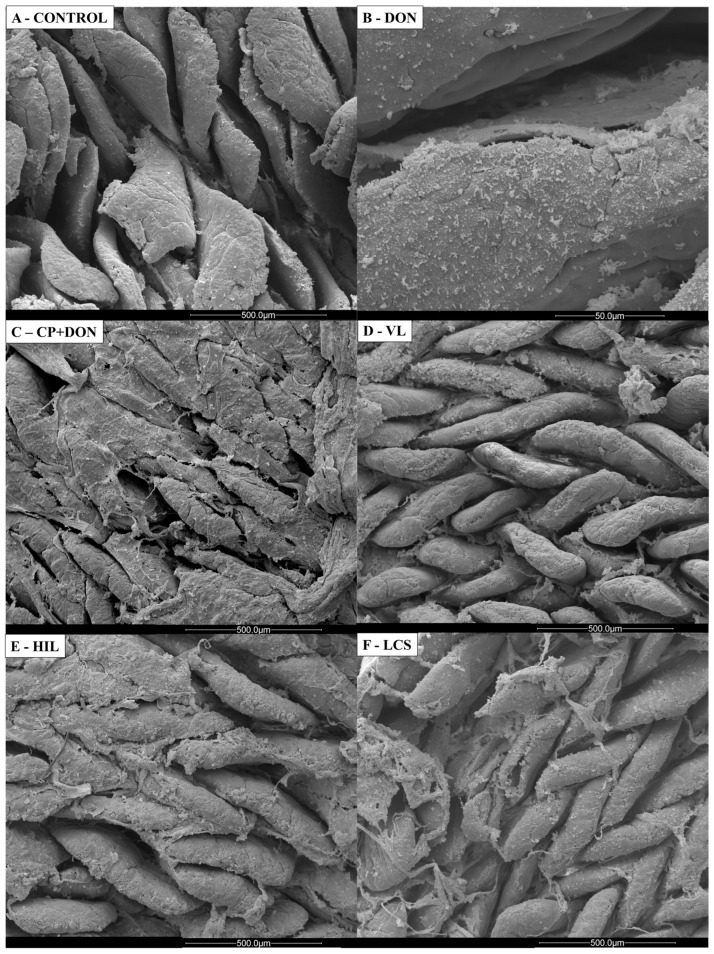
Scanning electron microscopy images illustrating the effect of *Lactobacillus* mixture in the jejunum after DON and CP challenge. (**A**) Control—normal villi morphology. (**B**) DON—enhanced presence of mucus compared to the control group. (**C**) CP + DON—enhanced presence of mucus. (**D**) Viable *Lactobacillus*—preserved villi integrity. (**E**) Heat-inactivated *Lactobacillus*—preserved villi morphology. (**F**) *Lactobacillus* culture supernatant—preserved villi morphology.

**Table 1 toxins-16-00046-t001:** Effect of *Lactobacillus* spp. mixture on villus height, crypt depth, villus: crypt ratio, microscopic lesion score, intraepithelial lymphocytes infiltration, and goblet cell count in the duodenum, jejunum, and ileum.

Treatment	Villus Height [µm]	Crypt Depth [µm]	Villi: Crypt Ratio	Microscopic Lesion Score	IEL	Goblet Cells
	**Duodenum**					
Control	1315.02 ± 189.85	145.91 ± 18.95	9.13 ± 1.78	7.50 ^b^ ± 1.22	NA	NA
DON	1216.70 ± 182.97	158.64 ± 19.96	7.84 ± 1.92	9.16 ^b^ ± 3.31	NA	NA
CP	1159.23 ± 202.50	133.11 ± 15.99	8.71 ± 1.19	11.83 ^a^ ± 5.81	NA	NA
CP + DON	1296.70 ± 258.16	148.60 ± 17.92	8.80 ± 1.93	15.40 ^a^ ± 3.71	NA	NA
VL	1299.06 ± 135.00	164.57 ± 43.96	8.22 ± 1.84	11.40 ^b^ ± 1.52	NA	NA
HIL	1352.86 ± 99.07	140.28 ± 13.89	9.76 ± 1.53	8.40 ^b^ ± 3.05	NA	NA
LCS	1180.26 ± 98.46	138.52 ± 14.33	8.55 ± 1.19	9.40 ^b^ ± 2.51	NA	NA
	**Jejunum**					
Control	970.52 ^a^ ± 159.33	133.87 ± 13.01	7.30 ^a^ ± 1.34	4.16 ^b^ ± 2.86	20.18 ^b^ ± 2.89	NA
DON	632.43 ^b^ ± 109.73	129.62 ± 12.69	4.89 ^b^ ± 0.83	8.16 ^a^ ± 1.47	25.68 ^b^ ± 2.66	NA
CP	840.82 ^b^ ± 232.02	126.74 ± 11.09	6.59 ^b^ ± 1.43	11.17 ^a^ ± 3.71	28.84 ^a^ ± 2.79	NA
CP + DON	789.68 ^b^ ± 77.20	127.23 ± 9.13	6.22 ^b^ ± 0.69	12.00 ^a^ ± 2.16	32.70 ^a^ ± 4.94	NA
VL	1075.19 ^a^ ± 154.24	122.39 ± 18.15	8.88 ^a^ ± 1.47	8.60 ^a^ ± 3.36	24.76 ^b^ ± 3.99	NA
HIL	961.53 ^a^ ± 225.59	131.70 ± 16.69	7.48 ^a^ ± 2.35	7.00 ^b^ ± 3.67	23.41 ^b^ ± 1.90	NA
LCS	901.30 ^a^ ± 94.33	113.33 ± 14.87	8.02^a^ ± 0.96	9.00 ^a^ ± 3.46	30.26 ^a^ ± 5.77	NA
	**Ileum**					
Control	750.29 ^a^ ± 49.07	123.43 ± 16.56	6.20 ± 1.21	4.50 ^c^ ± 1.87	18.98 ^b^ ± 1.07	66.97 ^c^ ± 5.87
DON	605.68 ^b^ ± 59.83	128.91 ± 25.98	4.86 ± 1.02	6.50 ^b^ ± 1.87	23.52 ^a^ ± 2.23	73.63 ^c^ ± 5.19
CP	671.15 ^b^ ± 107.38	118.87 ± 12.92	5.64 ± 0.62	8.67 ^a^ ± 3.93	26.11 ^a^ ± 5.50	87.34 ^b^ ± 11.32
CP + DON	713.88 ^a^ ± 86.42	115.98 ± 11.15	6.16 ± 0.58	9.00 ^a^ ± 1.87	26.75 ^a^ ± 2.47	88.54 ^b^ ± 9.36
VL	729.05 ^a^ ± 52.30	137.01 ± 13.66	5.34 ± 0.40	4.20 ^c^ ± 1.79	22.46 ^a^ ± 2.05	108.04 ^a^ ± 9.29
HIL	703.70 ^a^ ± 42.50	131.71 ± 10.05	5.35 ± 0.26	9.20 ^a^ ± 2.28	24.26 ^a^ ± 4.27	105.21 ^a^ ± 5.83
LCS	731.06 ^a^ ± 38.13	131.85 ± 16.41	5.59 ± 0.54	6.80 ^b^ ± 2.49	24.11 ^a^ ± 2.55	104.81 ^a^ ± 10.18

Values are the mean ± standard deviation. ANOVA followed by Scott–Knott multiple comparison test was used to determine significant differences among groups. ^a,b,c^ Different letters in the same column indicate a significant difference. Control—uncontaminated diet. DON (deoxynivalenol)—diet with DON 19.3 mg kg^−1^. CP (*Clostridium perfringens*)—uncontaminated diet + *C. perfringens* challenge. CP + DON—DON 19.3 mg kg^−1^ + *C. perfringens* challenge. LV—DON 19.3 mg kg^−1^ + *C. perfringens* challenge, supplemented with viable *Lactobacillus* spp. mixture. HIL—DON 19.3 mg kg^−1^ + *C. perfringens* challenge, supplemented with heat-inactivated *Lactobacillus* spp. mixture. LCS—DON 19.3 mg kg^−1^ + *C. perfringens* challenge, supplemented with *Lactobacillus* spp. mixture culture supernatant. NA—not analyzed.

**Table 2 toxins-16-00046-t002:** Effect of *Lactobacillus* spp. mixture on the oxidative status in the small intestine. Values are the mean ± standard deviation of the mean. ANOVA followed by Scott–Knott multiple comparison test was used to determine statistical differences among groups. ^a,b,c^ Different letters in the same column indicate a statistical difference.

Treatment	TBARS	NBT	GSH	ABTS	FRAP
**Jejunum**					
Control	0.04 ^b^ ± 0.03	6.67 ^b^ ± 3.19	2809.33 ^a^ ± 1215.38	0.70 ± 0.09	0.59 ^b^ ± 0.17
DON	0.08 ^a^ ± 0.05	7.56 ^b^ ± 2.30	1313.11 ^b^ ± 447.71	1.10 ± 0.18	0.75 ^b^ ± 0.25
CP	0.03 ^b^ ± 0.01	32.46 ^a^ ± 12.51	447.51 ^c^ ± 110.76	1.13 ± 0.44	1.65 ^a^ ± 0.56
CP + DON	0.05 ^a^ ± 0.01	24.51 ^a^ ± 13.54	458.97 ^c^ ± 20.38	1.11 ± 0.39	1.21 ^a^ ± 0.23
VL	0.03 ^b^ ± 0.01	32.85 ^a^ ± 27.97	593.57 ^c^ ± 75.59	1.00 ± 0.25	1.40 ^a^ ± 0.15
HIL	0.02 ^b^ ± 0.01	48.70 ^a^ ± 25.97	585.41 ^c^ ± 110.41	0.85 ± 0.25	1.55 ^a^ ± 0.33
LCS	0.03 ^b^ ± 0.01	25.55 ^a^ ± 16.27	400.20 ^c^ ± 107.88	1.37 ± 0.40	1.98 ^a^ ± 0.45
**Ileum**					
Control	0.02 ± 0.007	6.81 ^b^ ± 0.64	1568.72 ^a^ ± 731.15	0.81 ± 0.17	1.28 ± 0.94
DON	0.04 ± 0.03	6.48 ^b^ ± 3.46	1702.98 ^a^ ± 412.50	0.64 ± 0.19	0.96 ± 0.49
CP	0.02 ± 0.008	30.80 ^a^ ± 8.62	588.39 ^b^ ± 120.34	0.89 ± 0.36	1.78 ± 0.57
CP + DON	0.03 ± 0.02	44.17 ^a^ ± 14.05	556.43 ^b^ ± 111.85	0.75 ± 0.35	0.97 ± 0.21
VL	0.02 ± 0.01	39.59 ^a^ ± 14.33	639.80 ^b^ ± 250.44	0.88 ± 0.22	1.18 ± 0.30
HIL	0.03 ± 0.005	36.29 ^a^ ± 13.26	606.61 ^b^ ± 87.30	0.57 ± 0.34	1.41 ± 0.52
LCS	0.03 ± 0.009	26.03 ^a^ ± 17.26	663.84 ^b^ ± 137.50	0.59 ± 0.34	0.93 ± 0.23

Control—uncontaminated diet. DON (deoxynivalenol)—diet with DON 19.3 mg kg^−1^. CP (*Clostridium perfringens*)—uncontaminated diet + *C. perfringens* challenge. CP + DON—DON 19.3 mg kg^−1^ + *C. perfringens* challenge. LV—DON 19.3 mg kg^−1^ + *C. perfringens* challenge plus viable *Lactobacillus* spp. mixture. HIL—DON 19.3 mg kg^−1^ + *C. perfringens* challenge plus heat-inactivated *Lactobacillus* spp. mixture. LCS—DON 19.3 mg kg^−1^ + *C. perfringens* challenge plus *Lactobacillus* spp. mixture culture supernatant. Results are expressed as: TBARS (thiobarbituric acid reactive substances)—ΔOD A535−A532/mg of tissue; NBT (nitroblue tetrazolium)—OD/mg of protein; GSH (reduced glutathione)—nmol/mg of protein; ABTS (2,2′-azino-bis (3-ethylbenzothiazoline-6-sulphonic acid)—nmol Trolox Eq/mg of tissue; FRAP (ferric reducing antioxidant power)—nmol Trolox Eq/mg of tissue.

**Table 3 toxins-16-00046-t003:** Composition of the experimental diets.

Ingredients (g/kg)	Diet 1 0–6 Days	Diet 2 7–14 Days	Diet 3 15–23 Days
Corn	481.6	150	150
Soya bean meal (46% CP)	437	353.14	-
Soybean oil	32	54	54
Wheat	-	404.33	404.33
Fishmeal	-	-	353.14
Dicalcium phosphate	25	9.28	9.29
Limestone	1.5	13.17	13.16
Sodium chloride	6	5.03	5.04
Premix	5	5	5
L-lysine HCL	0.968	2.1	2.08
DL-methionine	3.36	3	2.99
L-threonine	0.38	0.98	0.98
Nutritional Levels			
Energy mcal/kg	2975	3050	3010
Protein (%)	24.27	23.31	25.61
Linoleic acid (%)	3.355	-	-
Calcium (%)	0.971	0.878	2.910
Phosphorus available (%)	0.463	0.310	1.563
Lysine dig (%)	1.307	1.256	1.418
Methionine dig (%)	0.646	0.600	0.869
Methionine + cistine dig (%)	0.967	0.929	1.176
Threonine dig (%)	0.863	0.829	0.931
Tryptophan dig (%)	0.277	0.271	0.190
Sodium (%)	0.225	0.218	0.451

Premix—Iron 8400.00 mg kg^−1^; Copper 3200.00 mg kg^−1^; Manganese 13.60 g kg^−1^; Zinc 10.80 g kg^−1^; Iodine 146.00 mg kg^−1^; Selenium 52.00 mg kg^−1^; Vitamin A 2,500,000.00 UI/kg; Vitamin D3 420,000.00 UI/kg; Vitamin E 6000.00 UI/kg; Vitamin K3 500.00 mg kg^−1^; Vitamin B1 500.00 mg kg^−1^; Vitamin B2 1600.00 mg kg^−1^; Niacin 7000.00 mg kg^−1^; Vitamin B6 900.00 mg kg^−1^; Folic acid 200.00 mg kg^−1^; Biotin 36.00 mg kg^−1^; Vitamin B12 16,000.00 mcg kg^−1^; Colin 80.00 g kg^−1^; Methionine 178.20 g kg^−1^.

**Table 4 toxins-16-00046-t004:** Mycotoxin contamination levels of the experimental diets used in the trial, as determined by HPLC/MS-MS.

Contamination Level [µg kg^−1^]			
Mycotoxin	1–6 Days Uncontaminated Diet	7–23 Days Uncontaminated Diet	7–23 Days Contaminated Diet
DON	<LOQ	200	19,309.4
AFB1	<LOQ	<LOQ	<LOQ
AFB2	<LOQ	<LOQ	<LOQ
AFG1	<LOQ	<LOQ	<LOQ
AFG2	<LOQ	<LOQ	<LOQ
FB1	252.9	<LOQ	<LOQ
FB2	<LOQ	<LOQ	<LOQ
ZEA	31.4	<LOQ	4878.7

LOQ: limit of quantification. DON: deoxynivalenol; AFB1: aflatoxin B1; AFB2: aflatoxin B2; AFG1: aflatoxin G1; AFG2: aflatoxin G2; FB1: fumonisin B1; FB2: fumonisin B2; ZEA: zearalenole. LOQ: DON, 200 µg kg^−1^; AFB1, AFB2, AFG1, AFG2, 1 µg kg^−1^; FB1, FB2, 125 µg kg^−1^; ZEA, 20 µg kg^−1^.

## Data Availability

All data are present in the article.

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
