# Peer review of "Modulation of Broiler Intestinal Changes Induced by Clostridium perfringens and Deoxynivalenol through Probiotic, Paraprobiotic, and Postbiotic Supplementation"

_toxins, 2024, doi:10.3390/toxins16010046_

Round 1

Reviewer 1 Report

Comments and Suggestions for Authors

The authors investigated the protective role of a mixture of Lactobacillus as probiotics, paraprobiotics, and post-probiotics in mitigating intestinal damage caused by DON and C. perfringens. This study contains some interesting findings and are valuable for the understanding of Lactobacillus spp. mitigating intestinal changes induced by DON and C. perfringens . However, the design and writing is not scientific enough. Also. many language errors throughout this manyscript.

It is unclear why it is important to study the combined effect of DON and CP on the intestinal damage.

It is unclear how did author select 19.3 mg/kg as the treatment concentration of DON.

There is no mechanism for Lactobacillus mitigating intestinal damage by DON and CP

Why the first figure is figure2 and the last figure is figure 1。 Also,the figure 1 is unnecessary, and it is better to list figure 1 in supplemental data.

Table 2: The layout of tables doesn’t fit rules of scientific journal. Also, why the first table is table 2. The tables and figures should be numbered as their appearance order.

Comments on the Quality of English Language

The english must be improved.

Author Response

December 28th 2023.

Toxins 

Dear reviewers,

Please accept our revised manuscript Modulation of Broiler Intestinal Changes Induced by Clostridium perfringens and Deoxynivalenol through Probiotic, Paraprobiotic, and Postbiotic Supplementation for consideration for publication in Toxins. We have gone through the valuable comments and considerably altered the manuscript accordingly.  

We have studied the comments carefully and made corrections, which we hope will meet with your approval. We strongly acknowledge your expertise and efforts.

The revised sections are marked in red in the manuscript. The main corrections in the manuscript and responses to your comments are as follows: 

Comments from Reviewer 1: 

The authors investigated the protective role of a mixture of Lactobacillus as probiotics, paraprobiotics, and post-probiotics in mitigating intestinal damage caused by DON and C. perfringens. This study contains some interesting findings and are valuable for the understanding of Lactobacillus spp. mitigating intestinal changes induced by DON and C. perfringens . However, the design and writing is not scientific enough. Also. many language errors throughout this manyscript.

  1. It is unclear why it is important to study the combined effect of DON and CP on the intestinal damage.

Answer: In lines 46-57 the importance to study the combined effects of DON and CP is exposed.

  1. It is unclear how did author select 19.3 mg/kg as the treatment concentration of DON.

Answer: We used a high concentration of DON to be successful in inducing lesions and thus have an effective evaluation of the protective effect of the Lactobacillus pool.

  1. There is no mechanism for Lactobacillus mitigating intestinal damage by DON and CP

Answer: In lines 288-298 a mechanism for Lactobacillus spp. mixture mitigating intestinal damage by DON and CP is proposed.

  1. Why the first figure is figure2 and the last figure is figure 1. Also,the figure 1 is unnecessary, and it is better to list figure 1 in supplemental data.

Answer: we are grateful for the suggestion that is accepted.

  1. Table 2: The layout of tables doesn’t fit rules of scientific journal. Also, why the first table is table 2. The tables and figures should be numbered as their appearance order.

Answer: the layout is revised as well as the figures numbers.

Reviewer 2 Report

Comments and Suggestions for Authors

Author Response

December 28th 2023.

Toxins 

Dear reviewers,

Please accept our revised manuscript Modulation of Broiler Intestinal Changes Induced by Clostridium perfringens and Deoxynivalenol through Probiotic, Paraprobiotic, and Postbiotic Supplementation for consideration for publication in Toxins. We have gone through the valuable comments and considerably altered the manuscript accordingly.  

We have studied the comments carefully and made corrections, which we hope will meet with your approval. We strongly acknowledge your expertise and efforts.

The revised sections are marked in red in the manuscript. The main corrections in the manuscript and responses to your comments are as follows: 

Comments from Reviewer 2:

  1. Whether the two words are repeated?

Answer: In the lines 12 and 13 the words morphology and morphometrys have different meanings. Morphology refers to the intestinal villi form and morphometry refers to measureable variables such as villi height, crypt depth and villi:crypt ratio.

  1. The order described here should be consistent with the order of the results.

Answer: The sentence has been revised. Line 13.

  1. Use", " to separate keywords.

Answer: the issue was corrected. Line 26.

  1. This word is not appropriate, please change it.

Answer: Resurgence was replaced by reemergence. Line 34.

  1. The subject is not clear, it is recommended to split into two sentences.

Answer: The sentence has been revised. Lines 35-36.

  1. The effect of probiotics was mentioned in the results and it is suggested to change the title.

Answer: The sentence has been revised. Line 76.

  1. Please describe in detail in which picture you observed these changes.

Answer: The sentence has been revised. Line 80-81.

  1. If it is possible, please add intestinal mucosal pictures of VL, HIL or LCS group.

Answer: A picture from VL group was added. Line 83.

  1. Leave a little gap between the pictures and label the group name above each pictures.

Answer: Done as requested. Line 148.

  1. Where is the H diagram?

Answer: The mistake was corrected in the text. Line 169.

  1. Abbreviations that appear for the first time should preceded by the full name.

Answer: Done as requested. Lines 175-177.

  1. Why not make a statistical graph to make the data more intuitive?

Answer: Considering that most of the results showed no significant difference among treatments we have opted to describe these results in this section.

  1. The protective role of Lactobacillus in immunity was not mentioned in the discussion.

Answer: It was not mentionated because there was no statistical significance among treatments (p= 0.08).

  1. Please add experimental results or relevant references here.

Answer: New information is added. Lines 292-294.

  1. Ways to improve the use of Lactobacillus probiotics can be suggested at the end of the discussion, or to look forward to other aspects.

Answer: The sentence has been revised. Lines 300-306.

  1. Please supplement the criteria or calculation method of macroscopic lesion score.

Answer: The sentence has been revised. Lines 414-416.

Reviewer 3 Report

Comments and Suggestions for Authors

OVERALL EVALUATION

The article deals with an important aspect of chicken farming, but has many limitations that prevent it from being accepted.

The article may be resubmitted once the weak points have been clarified.

MAJOR REMARKS

Experimental unit has not been clearly defined: were the poultry reared in single cages ?

How was the gut lesion score calculated ?

Furthermore, feed intake has not been reported and this is an important topic, because DON usually reduces feed intake.

The amount of Lactobacillus mixture (CFU/ml)  given to animals must be reported by Authors

The chemical composition of the diet must be reported.

Line 316-317. The methods used for the determination of mycotoxins in feed must be reported.

Table 3. The unit of measure of the concentration of molecules has not been reported.

 MINOR REMARKS

Line 14.  Please correct the typing mistake “the potential of a of”.

Table 2. Please put “µm” into brackets.

Author Response

December 28th 2023.

Toxins 

Dear reviewers,

Please accept our revised manuscript Modulation of Broiler Intestinal Changes Induced by Clostridium perfringens and Deoxynivalenol through Probiotic, Paraprobiotic, and Postbiotic Supplementation for consideration for publication in Toxins. We have gone through the valuable comments and considerably altered the manuscript accordingly.  

We have studied the comments carefully and made corrections, which we hope will meet with your approval. We strongly acknowledge your expertise and efforts.

The revised sections are marked in red in the manuscript. The main corrections in the manuscript and responses to your comments are as follows: 

Comments from Reviewer 3:

OVERALL EVALUATION

The article deals with an important aspect of chicken farming, but has many  limitations that prevent it from being accepted.

The article may be resubmitted once the weak points have been clarified.

MAJOR REMARKS

  1. Experimental unit has not been clearly defined: were the poultry reared in single cages?

Answer: the animals were reared in groups of six and each animal was considered one experimental unit. Line 470-471.

  1. How was the gut lesion score calculated ?

Answer: For the macroscopic gut evaluation a score ranging from 0-5 was used to each animal (Cravens et al. 2013). For the microscopic lesion score the methodology described by Terciolo et al. (2019) was applied. Briefly  the lesion score was obtained by multiplying the severity factor with the extent of the lesion. The extent of each lesion (intensity or frequency) was evaluated and scored as: 0, no lesion; 1, low extent (25 % of the intestinal section affected); 2, intermediate extent (50 % of the intestinal section affected); 3, large extent (75 % of the intestinal section affected).

  1. Furthermore, feed intake has not been reported and this is an important topic, because DON usually reduces feed intake.

Answer: During the experimental period the animals did not refuse the diet and a reduction in the feed intake was not observed. We did not evaluate the feed intake during the experimental period because the evaluation of the zootechnical performance was not a goal in this study. In addition, the reduced number of animals per treatment is a limitation to perform statistical analysis of this result, since each animal was considered one experimental unit.

  1. The amount of Lactobacillus mixture (CFU/ml) given to animals must be reported by Authors.

Answer: The information is available in lines 392-393.

  1. The chemical composition of the diet must be reported.

Answer: Done as requested, Table 3. Line 365.

  1. Line 316-317. The methods used for the determination of mycotoxins in feed must be reported.

Answer: The mycotoxin’s levels in feed were determinated by high performance liquid chromatography. This information is available on Table 4, lines 371-375.

  1. Table 3. The unit of measure of the concentration of molecules has not been

Answer: Table 3 was renumbered as Table 2, the unit of measure of each measured molecule is reported in the legend. Lines 196-200: Results are expressed as: TBARS (thiobarbituric acid reactive substances) – ΔOD A535-A532/mg of tissue; NBT (nitroblue tetrazolium) OD/mg of protein; GSH (reduced glutathione) - nmol/mg of protein; ABTS (2,2′-azino-bis (3-ethylbenzothiazoline-6-sulphonic acid) – nmol Trolox Eq/mg of tissue; FRAP (ferric-reducing antioxidant power) - nmol Trolox Eq/mg of tissue.

 MINOR REMARKS

  1. Line 14. Please correct the typing mistake “the potential of a of”.

Answer: The mistake was corrected. Line 14.

  1. Table 2. Please put “µm” into brackets.

Answer: Done as requested. Line 121.

References

CRAVENS, R. L.; GOSS, G. R.; CHI, F.; DE BOER, E. D.; DAVIS, S. W.; HENDRIX, S. M.; RICHARDSON, J. A.; JOHNSTON, S. L. The effects of necrotic enteritis, aflatoxin B1, and virginiamycin on growth performance, necrotic enteritis lesion scores, and mortality in young broilers. Poultry Science, v. 92, n. 8, p. 1997-2004, 2013.

TERCIOLO, C.; BRACARENSE, A. P.; SOUTO, P. C. M. C.; COSSALTER, A.-M.; DOPAVOGUI, L.; LOISEAU, N.; OLIVEIRA, C. A. F.; PINTON, P.; OSWALD, I. P. Fumonisins at Doses below EU Regulatory Limits Induce Histological Alterations in Piglets. Toxins, v. 11, n. 9, p. 548,  2019.

Reviewer 4 Report

Comments and Suggestions for Authors

In Materials and Methods under 4.3. Diets it is mentioned that mycotoxin analyses were performed in a private laboratory. 

However, is missing  the description of the chromatographic method, mobile phase composition, column, gradient, oven temperature, which detector ? These data should be mentioned in a subchapter under material and methods. 

The paragraph between lines 199- 202 of the Discussion should be moved to the Introduction. 

There is no conclusion chapter, just a sentence between lines 284-286. 

Comments on the Quality of English Language

Moderate editing of English required

Author Response

December 28th 2023.

Toxins 

Dear reviewers,

Please accept our revised manuscript Modulation of Broiler Intestinal Changes Induced by Clostridium perfringens and Deoxynivalenol through Probiotic, Paraprobiotic, and Postbiotic Supplementation for consideration for publication in Toxins. We have gone through the valuable comments and considerably altered the manuscript accordingly.  

We have studied the comments carefully and made corrections, which we hope will meet with your approval. We strongly acknowledge your expertise and efforts.

The revised sections are marked in red in the manuscript. The main corrections in the manuscript and responses to your comments are as follows: 

Comments from Reviewer 4:

  1. In Materials and Methods under 4.3. Diets it is mentioned that mycotoxin analyses were performed in a private laboratory. However, is missing the description of the chromatographic method, mobile phase composition, column, gradient, oven temperature, which detector ? These data should be mentioned in a subchapter under material and methods.

Answer: The sentence has been revised. Line 334-337. We would like to clarify that in most of the published papers concerning mycotoxins effects in animal health a detailed description of mycotoxin methodology detection is not usual. The Lamic laboratory is a well-recognized laboratory in Brazil to mycotoxin detection. More than 1.4 million samples have been analyzed in the last 25 years at Lamic.

“A blend (standard diet + DON) was prepared at Universidade Estadual de Londrina facilities using a commercial feed mixer. The diets were sent to Lamic laboratory (Santa Maria – RS/Brazil), where the mycotoxins levels were assessed using the HPLC/MS-MS method. The control diet showed lower levels of DON than the limit of quantification established (200 μg/kg with a stated accuracy of 101%).”

  1. The paragraph between lines 199- 202 of the Discussion should be moved to the Introduction.

Answer: We appreciate your suggestion, the paragraph has been revised.

  1. There is no conclusion chapter, just a sentence between lines 284-286.

Answer: The sentence has been revised. Line 298.

Round 2

Reviewer 1 Report

Comments and Suggestions for Authors

All questions has been addressed. The revised version is significantly improved. 

Reviewer 2 Report

Comments and Suggestions for Authors

accept

Reviewer 4 Report

Comments and Suggestions for Authors

I have no other additional requirements

Comments on the Quality of English Language

Minor editing of English required